# Deregulation of the Purine Pathway in Pre-Transplant Liver Biopsies Is Associated with Graft Function and Survival after Transplantation

**DOI:** 10.3390/jcm9030711

**Published:** 2020-03-05

**Authors:** Jin Xu, Mohammad Hassan-Ally, Ana María Casas-Ferreira, Tommi Suvitaival, Yun Ma, Hector Vilca-Melendez, Mohamed Rela, Nigel Heaton, Wayel Jassem, Cristina Legido-Quigley

**Affiliations:** 1Institute of Pharmaceutical Science, Faculty of Life Sciences & Medicine, King’s College London, London SE1 9NH, UK; jin.xu@kcl.ac.uk (J.X.); mha101@outlook.com (M.H.-A.); anacasas@usal.es (A.M.C.-F.); 2Department of Analytical Chemistry, Nutrition and Food Science, University of Salamanca, 37008 Salamanca, Spain; 3Steno Diabetes Center Copenhagen, DK-2800 Gentofte, Denmark; tommi.raimo.leo.suvitaival@regionh.dk; 4Institute of Liver Studies, King’s College Hospital, King’s College London, London SE5 9RS, UK; yun.ma@kcl.ac.uk (Y.M.); hector.vilca_melendez@kcl.ac.uk (H.V.-M.); mohamed.rela@gmail.com (M.R.); nigel.heaton@nhs.net (N.H.)

**Keywords:** graft function, survival, liver transplantation, metabolomics

## Abstract

The current shortage of livers for transplantation has increased the use of marginal organs sourced from donation after circulatory death (DCD). However, these organs have a higher incidence of graft failure, and pre-transplant biomarkers which predict graft function and survival remain limited. Here, we aimed to find biomarkers of liver function before transplantation to allow better clinical evaluation. Matched pre- and post-transplant liver biopsies from DCD (*n* = 24) and donation after brain death (DBD, *n* = 70) were collected. Liver biopsies were analysed using mass spectroscopy molecular phenotyping. Discrimination analysis was used to parse metabolites differentiated between the two groups. Five metabolites in the purine pathway were investigated. Of these, the ratios of the levels of four metabolites to those of urate differed between DBD and DCD biopsies at the pre-transplantation stage (*q* < 0.05). The ratios of Adenosine monophosphate (AMP) and adenine levels to those of urate also differed in biopsies from recipients experiencing early graft function (EGF) (*q* < 0.05) compared to those of recipients experiencing early allograft dysfunction (EAD). Using random forest, a panel consisting of alanine aminotransferase (ALT) and the ratios of AMP, adenine, and hypoxanthine levels to urate levels predicted EGF with area under the curve (AUC) of 0.84 (95% CI (0.71, 0.97)). Survival analysis revealed that the metabolite classifier could stratify six-year survival outcomes (*p* = 0.0073). At the pre-transplantation stage, a panel composed of purine metabolites and ALT could improve the prediction of EGF and survival.

## 1. Introduction

There is an increasing need for organ transplantation, but the number of organs available remains insufficient [1,2]. This is reflected by the number of people registered in the Organ Donor Register (ODR) in the UK, which decreased from 2018 to 2019 [3], while in the same period, the number of patients on the active transplant list increased by 20%, reaching the number of 432 [3]. This stark surge in the demand for liver transplants (LT) is attributable to the global incidence of alcohol-related fatty liver disease, cirrhosis and hepatitis [4].

To safeguard patients, pre-transplant donor screening is used to determine the probability of successful liver transplant. Optimal donors’ parameters in the case of donation after circulatory death (DCD) include age (<60 years), weight (<100 kg), intensive care stay (<5 days), functional warm ischaemia time (fWIT, <20 min), cold ischaemia time (<8 h) and steatosis (<10%) [5]. These values have resulted in up to 20% of donation-after-brain-death (DBD) organs not meeting the clinical criteria [6] and a 78% increase in the discard rate of DCD livers [7]. The application of these criteria can result in a number of otherwise transplantable organs being discarded [8]. Therefore, identifying specific pre-transplantation markers of liver damage could assist in expanding the pool of transplantable livers.

Currently, the standard assessment of liver dysfunction is carried out using liver function tests that evaluate the concentrations of liver enzymes such as alkaline phosphatase (ALP), alanine aminotransferase (ALT), aspartate aminotransferase (AST) and gamma-glutamyl transferase (GGT) [9,10]. However, such tests lack sensitivity and specificity and can be affected by patient factors such as genetics, medicines and other non-associated diseases [11,12,13,14]. Thus far, transcriptomics and genomics have been used to discover biomarkers in live pathophysiology [15]. Metabolomics has also been employed to decipher metabolic fluxes in liver disease [16]. A systematic review on the use of metabolomics to discover liver biomarkers for transplantation outcomes in liver tissue biopsies highlighted promising results [17]. These first studies identified lipid molecules [18,19,20], tryptophan, kynurenine and S-adenosylmethionine as liver biomarkers [21,22].

The objective of this study was to employ a molecular phenotyping approach to investigate, at both pre- and post-transplantation, hundreds of polar metabolites in the hepatic tissue from two distinct donor types, viz., DBD and DCD donors. Following this, the association between metabolites that were different between these donor types and clinical outcomes, viz., early allograft dysfunction (EAD) and early graft function (EGF), were investigated. Then, prediction of EGF was calculated, and survival analysis based on metabolites and clinical variables was performed. The study workflow is illustrated in Appendix A.

## 2. Materials and Methods

### 2.1. Patients and Samples

This study received prior approval from the ethics committee at King’s College Hospital (ethical approval number 09/H0802/100), and informed consent was obtained from all subjects. The methods were carried out in accordance to the ethical guidelines of the 1975 Declaration of Helsinki, and no donor organs were obtained from executed prisoners or other institutionalized persons.

Overall 94 Tru-Cut tissue biopsies were obtained from the left lobe of livers pre- and post-transplantation. The first (pre-transplant) biopsy was taken at the end of cold preservation, prior to implantation, and the second (post-transplant) biopsy was obtained approximately 1 h after graft reperfusion. A separate biopsy was obtained for histopathological evaluation of donor steatosis. Biopsies were immediately snap-frozen in liquid nitrogen and stored at −80 °C until extraction for LC–MS analysis. In all procedures, liver allografts were flash-cooled and perfused with University of Wisconsin preservation solution until the time of transplantation.

The study included two types of adult donors: DBD (*n* = 35) and DCD (*n* = 12). A wide spectrum of donor clinical data were collected for comparison among groups and for correlation with metabolite levels. In the DCD group, functional WIT (fWIT) was calculated from the time when systolic blood pressure was below 50 mmHg to the time of aortic cannulation. All recipients were patients with stable chronic liver disease who did not require hospitalization prior to transplantation. They also presented with a similar severity of liver disease, represented by scores assessed using the Model for End-Stage Liver Disease (MELD) at time of listing for LT. DCD donor liver grafts were randomly selected from transplants performed from August 2011 to August 2014, and all graft were matched with DBD grafts performed in the same period. After transplantation, all patients received immunosuppressive therapy with tacrolimus and prednisolone. Graft performance was assessed based on serum AST, serum bilirubin and international normalised ratio (INR) levels after transplantation [23]. According to graft performance, recipients were classified into two groups, i.e., patients showing EAD (*n* = 10) and those showing EGF (*n* = 37). The survival data were collected for 34 recipients from the time of transplantation (between 2011 and 2014) till April 2019. The relevant donor and recipient details are listed in Table 1.

### 2.2. Sample Treatment

Sample preparation for all 94 biopsies followed our previously published method [24]. We transferred 100 µL of the lower aqueous phase from all samples to clean vials for further analysis. The samples were kept in the chamber at a temperature of 4 °C, and the injection volume was 5 µL, with full-loop function (20 µL loop size). Chromatographic and spectrometric conditions for the analysis of polar metabolites were according to a published protocol [25]. Quality controls (QC) were run every 8 samples in random order.

### 2.3. Statistics

All data were processed within the “XCMS” package in “R Studio” (version 1.0.153), and multivariate analyses were conducted in both “R Studio” and “SIMCA” (version 14, MKS Umetrics AB, Umeå, Sweden). Multivariate analysis included pre- and post-transplant matched samples *n* = 94 (DBD *n* = 70, DCD *n* = 24) and 17 QCs. Principle component analysis (PCA) was carried out to detect outlier(s) and to examine the distribution of QCs. All pre- transplant data were then divided into a training dataset (DBD *n* = 30, DCD *n* = 5) and a test dataset (DBD *n* = 5, DCD *n* = 7). An orthogonal projections to latent structures discriminant analysis (OPLS-DA) model was built based on the training dataset to examine the profiling of pre- transplant samples in DBD and DCD groups. The test dataset was utilised to assess the prediction ability of the built model. S-plot derived from the OPLS-DA model was then applied to select features based on covariance P1 and correlation P (corr) values (P1 > 0.1, P (corr) > 0.4 and P1 < −0.1, P (corr) < −0.4).

Metabolic features based on the LC–MS data were measured using Waters MassLynx software (Waters Corporation, Milford, MA, USA). Feature concentrations were expressed as ratios of peak areas to internal standards’ peak areas. The identification was performed by using metabolites mass to search against in-house and public metabolite databases [26,27,28]. The metabolites’ structure and fragmentation patterns in the MS2 data were studied by comparison with those of pure standard molecules.

To compare between DBD and DCD as well as between EAD and EGF groups at pre- and post-transplantation stages, levels of the identified metabolites and their ratios to the levels of another metabolite of the selected ones were explored and examined with univariate non-parametric Mann–Whitney test (2-sided) and Benjamini–Hochberg test. The ratios of selected metabolite in normal (no steatotic, *n* = 21) and steatotic (mild and moderate steatotic, *n* = 26) groups were also investigated. Post-hoc power calculation was performed for EGF (*n* = 37) and EAD (*n* = 10) participants using the values of metabolite ratios in “Gpower3.1”.

Furthermore, random forest machine learning and receiver operating characteristic (ROC) were applied to choose the best predictors of EGF from the above selected ratios. Three ROC curves were determined: the highest possible area under the curve (AUC) with the combination of either clinical variables or metabolites (or their ratios) and the highest AUC with the combination of metabolites and clinical variables (package “caret”, “randomForest”, “pROC” and “ggplot2” in R studio). To follow this, correlation analyses between annotated metabolites and clinical features (serum AST, bilirubin, GGT) were conducted. Calculations were conducted in SPSS 23 (IBM, Armonk, NY, USA). Figures were plotted in GraphPad Prism 6 (GraphPad, La Jolla, CA, USA).

Metabolite ratios, clinical variables and the type of liver donor information were compared for their predictive power of survival. Two logistic regression models were fitted to make predictions based on metabolite ratios and clinical variables, respectively. Third, the group variable was used for the predictions as such. Participants were stratified into two equal-sized groups based on each of the three prediction models, and survival of these strata were compared with Kaplan–Maier curves. (package “survival”, “survminer” and “ggplot2” in R 3.4.2).

## 3. Results

### 3.1. Clinical Outcomes 

Demographics of all 94 patients in both groups are presented in Table 1. There were no significant differences between DBD and DCD groups in age, EAD/EGF, liver enzyme levels, hepatic steatosis or serum bilirubin levels. Differences were observed in recipient ages (*p* < 0.05) between groups.

### 3.2. Multivariate Model and Feature Selection

The unit variance (UV) scaled dataset was first inspected for detection of outlier(s). Next, the comparison between DBD and DCD samples at the pre-transplant stage was performed. An OPLS-DA model was built with a training dataset (DBD *n* = 30, DCD *n* = 5), and the model was tested with a test dataset (DBD *n* = 5, DCD *n* = 7). As shown in the misclassification table, the test samples in the DBD group could be predicted with 100% accuracy, while the DCD samples were predicted with 85.71% accuracy (Appendix A).

In order to identify which metabolic features were the strongest discriminators between DBD and DCD at pre-transplant, an S-plot (Appendix A) derived from the OPLS-DA model was used to select 12 features on the criteria stated in the Section 2. From the 12 selected features, 5 metabolites were annotated (Appendix A).

Five features were identified as purines at pre-transplant, and their levels in DBD and DCD were represented as bar plots in Appendix A. Additionally, jittered scatterplots representing the ratios of the levels of four purines to those of urate, illustrated in Figure 1, were plotted. At the pre-transplant stage, the ratios AMP/urate, adenosine/urate, adenine/urate and hypoxanthine/urate were significantly higher in the DBD group compared to the DCD one (*q* < 0.001). Moreover, the scatter plots showed that the ratios AMP/urate and adenine/urate were higher in the EGF group compared to the EAD group. The Mann–Whitney test confirmed that the mean ratios of adenine/urate and AMP/urate were significantly different between EAD and EGF (*q* < 0.05). Adenosine/urate and hypoxanthine/urate showed no significant difference between EAD and EGF groups.

At the post-transplant stage, the ratios AMP/urate, adenine/urate and hypoxanthine/urate (*q* < 0.05) were significantly higher in the DBD group compared to the DCD group. Additionally, the scatter plot illustrated that the ratios AMP/urate, adenosine/urate and adenine/urate were elevated in the EGF group compared to the EAD one (*q* < 0.05).

The comparison of metabolite ratio levels between normal and steatotic groups revealed no significant difference (Appendix A). In addition, post-hoc power was determined to asses this study, and the result was 77% power to detect differences between the EGF and EAD groups.

### 3.3. Random Forest with Metabolites and Clinical Variables

Machine learning was applied to identify variables to acting as classifiers between the EAD and EGF groups. From the included variables (AMP/urate, adenine/urate, hypoxanthine/urate, adenosine/urate, ALT, bilirubin, AST, GGT, steatosis status and donor age), high-importance scores for EGF were observed for the ratios of AMP/urate, adenine/urate, and hypoxanthine/urate and for ALT (Figure 2A). The prediction ability of purine ratios and ALT at pre-transplant was evaluated with ROC analysis. The accuracy, area under the curve (AUC), sensitivity and specificity for individual metabolites, enzymes and their various combinations in predicting EGF are listed in Table 2. The combination of the three ratios between purine and urate levels and ALT showed reliable prediction ability with high AUC, while the combination of four ratios between purine and urate levels demonstrated relatively higher accuracy, specificity and sensitivity (Figure 2B). Using random forest, a panel composed of ALT and the ratios of AMP, adenine and hypoxanthine levels to urate levels predicted EGF, with AUC of 0.84 (95% CI (0.71, 0.97)). In comparison, an AUC of 0.71 (95% CI (0.52, 0.90)) was achieved using the clinical parameters.

In order to investigate whether the levels of liver enzymes were associated with those of the analyzed purines, partial correlation analysis was performed. Purine relative amounts in pre- and post-transplant samples, together with serum AST, bilirubin and GGT in donors on the day of operation (day 0) and in recipients on the day after operation (day 1) were included for correlation analyses. In Table 3, the only significant correlation (*q* < 0.05) was observed between hypoxanthine and serum bilirubin after Benjamini–Hochberg correction.

### 3.4. Survival Analysis Based on Purines, Clinical Variables and Donation Groups

The purine ratio predictor (AMP/urate, adenine/urate, hypoxanthine/urate, adenosine/urate) stratified the participants: all five deaths occurred in the <50% strata, in which the metabolites predicted a lower chance of survival (Figure 3A; *p* = 0.073). For the clinical predictor (ALT, bilirubin, AST, GGT, steatosis status and donor age), three out of the five deaths occurred in the <50% strata, which indicates no significant prediction (Figure 3B; *p* = 0.54). Similarly, three out of five deaths occurred in the DBD group, and group class could not predict survival (Figure 3C; *p* = 0.15).

## 4. Discussion

The five metabolites that were highly correlated to DCD are generated in the purine metabolism pathway (Figure 4) [29,30]. Metabolites in the purine pathway have a myriad of functions and are important in regulating inflammation [31] and oxidative injury and as markers of cell death. In liver tissue undergoing cold and warm ischemia, the dysregulation of their levels could be related to energy, inflammation and ischemic tissue damage [32,33,34].

Purines can act as physiological regulators of leucocyte function [35], but to be functional they must be released in the appropriate microenvironment following stimuli [36]. It is thought that liver inflammation is due to a cascade of inflammatory events that occur mainly in donors after brain death [37]. On the other hand, our studies have shown that DCD grafts undergo low inflammation and increased hepatocellular damage due to warm ischaemia time [37,38].

In this study, AMP and adenine were found to be critically decreased in DCD. Studies have shown that adenine and AMP have a protective function during ischaemia [39]. Roy et al. found that, in addition to being mediators for graft recovery, high levels of AMP during ischaemia when oxygen is low indicate that ATP is still being generated [40,41]. This might explain why DBD allografts and the EGF group showed increased levels of both metabolites, as a higher energy reserve could improve post-transplant graft function [42]. 

AMP is also known to be protective during inflammation. It is generated from ATP and ADP by ectoapyrase (CD39) and released at the site of vascular injury when platelets aggregate to promote endothelial barrier function during inflammation [30]. Michael et al. found that overexpression of CD39 and hence increased AMP production conferred protection in both warm and cold hepatic ischaemia [43].

Adenine has been employed as a substrate to promote recovery. It has been shown that cells dying as a result of ischaemia undergo lysis to release adenine [44]. Kartha et al. demonstrated in vitro that adenine nucleotides accelerated structural and functional recovery in epithelial cells [45]. This would suggest that DBD liver allografts (Appendix A), with elevated levels of adenine at pre-transplant, may recover more rapidly.

Although not significantly elevated at pre-transplant, the levels of adenosine and hypoxanthine showed the same trend as those of AMP and adenine. Wyatt et al. found that a solution containing hypoxanthine and adenosine enhanced functional organ recovery after ischaemia/reperfusion (I/R) injury in dogs [46].

Increased levels of urate were observed in DCD livers pre-transplant (*q* <  0.001) (Appendix A). In humans, urate is the final product of purine metabolism [47]. In an experiment conducted by Matthew et al., in which hepatic ischaemia was induced for 30 min followed by 60 min of reperfusion, urate levels increased by over 300% after ischaemia and by 600% during the first 30 min of reperfusion [48]. Clear differences were revealed between DBD and DCD groups, as well as between EGF and EAD groups when the ratios of purines levels to urate levels were investigated (Figure 1). Epidemiological studies have also suggested that during I/R injury, urate levels are increased [49]. DCD allografts are more prone to IRI due to their exposure to a period of warm ischaemia [50]. 

We wanted to calculate the prediction ability of classifiers including purines for outcomes of EGF and longer-term survival. The model for EGF revealed that the diagnostic potential of combining the three ratios (AMP/urate, adenine/urate and hypoxanthine/urate) and ALT was the highest for EGF prediction, reaching 84% with a confidence interval of 71% to 96%. While higher accuracies were observed when purine levels were combined with known risks and enzyme markers (Table 2), the confidence interval shows that our study needs replication in a bigger cohort. Also, considering the average post-hoc power of 77% to distinguish EGF from EAD, at least 52 samples (EGF = 41, EAD = 11, with the same sample ratios used in this study) are needed to increase the power to 80%. The alterations we observed in regard to metabolite ratios were not related to the steatosis status, as no significant difference was observed between the normal and the steatotic groups. The survival analysis revealed that metabolite ratios were the best predictor of survival, compared to the other classifiers, i.e., clinical variables and the type of liver donor. Again, metabolite ratios better predicted deaths in the small dataset in comparison to clinical variables and the donor type. These preliminary results indicate that purine ratios may be useful in predicting prognosis, in addition to clinical profiles or donor graft types. However, we reiterate that the small number of patients in this study and samples that were limited to biopsies from operations conducted in one centre warrant that validation should be performed through a multicentre trial assessing early graft function. Also, a limitation of this study is that the first biopsy was taken before reperfusion, and for optimal results, a biopsy from the donor should also be included in the study design. To translate to the clinics and minimise the turnaround time of this panel (TAT), this test could be performed intraoperation, using available technology like rapid evaporative ionisation mass spectrometry (REIMS).

## 5. Conclusions

In this study, the combination of AMP/urate, adenine/urate, hypoxanthine/urate and ALT proved to have higher prediction ability compared to a combination of conventional liver function and risk markers. This study proposes a panel of small molecules at pre-transplantation that can aid in testing liver tissue quality for liver transplantation.

## Figures and Tables

**Figure 1 jcm-09-00711-f001:**
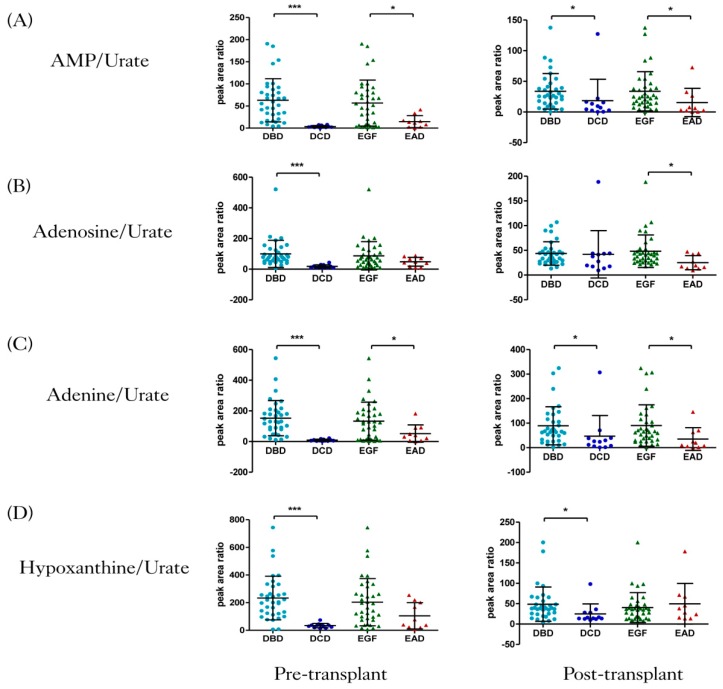
Jittered scatter plots of four ratios of metabolites’ levels in four groups at two transplant stages. (**A**) Adenosine monophosphate (AMP)/urate, (**B**) adenosine/urate, (**C**) adenine/urate, and (**D**) hypoxanthine/urate. AMP, adenosine monophosphate. Results are presented as mean ± SD, *p*-value was derived from Mann–Whitney tests, followed by Benjamini–Hochberg false discovery rate (FDR) correction (* *q* < 0.05, *** *q* < 0.001). DBD, donation after brain death; DCD, donation after circulatory death; EGF, early graft function; EAD, early allograft dysfunction.

**Figure 2 jcm-09-00711-f002:**
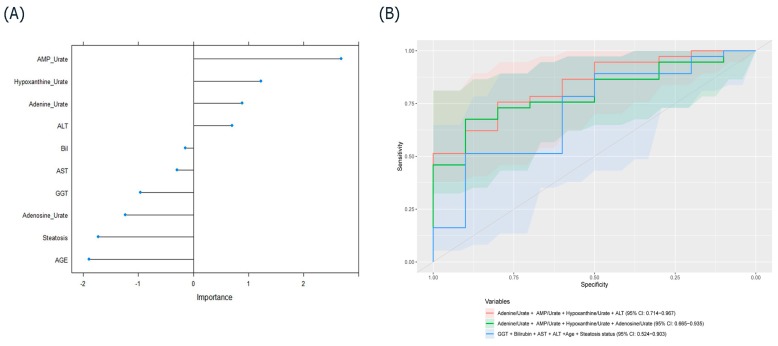
(**A**) Variable-importance plot derived from the random forest model. (**B**) Receiver operating characteristic (ROC) curve prediction of EGF based on the highest areas under the curve (AUC) with the combination of either clinical variables or metabolite ratios and the combination of metabolites and clinical variables. ALT, alanine aminotransferase; BiL, bilirubin; AST, aspartate aminotransferase; GGT, gamma-glutamyl transferase.

**Figure 3 jcm-09-00711-f003:**
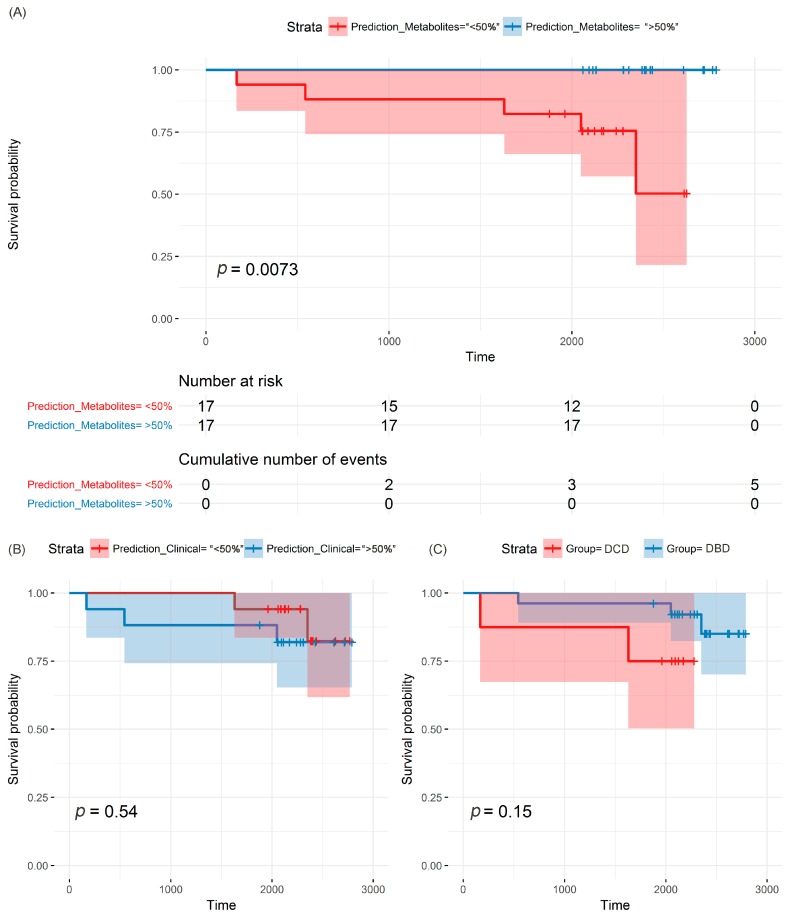
Kaplan–Meier plots of patients’ survival estimates, using three different types of predictors: (**A**) metabolite ratios, (**B**) clinical variables, (**C**) donation group.

**Figure 4 jcm-09-00711-f004:**
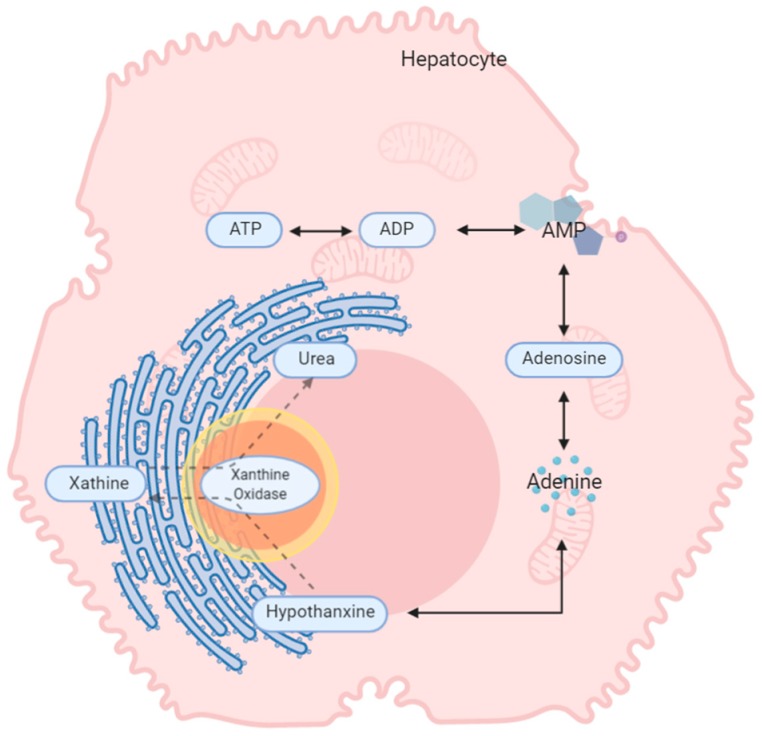
The proposed metabolic changes taking place in explanted liver. During energy production, phosphate groups are sequentially hydrolysed from ATP, creating ADP and then AMP. From AMP, the other metabolites are generated via a number of catabolic pathways. ATP, adenosine triphosphate; ADP, adenosine diphosphate.

**Table 1 jcm-09-00711-t001:** Demographic characteristics and clinical data of the 94 subjects involved in this study.

**Donor**		**DBD (*n* = 35)**	**DCD (*n* = 12)**	***p*-Value ^b^**
Age (years)		53 (25–82)	56 (35–76)	0.526
Gender (female/male)		19/16	6/6	1
Hepatic steatosis	No	14	7	
	Mild (<30%)	18	3	0.305
	Moderate (30–60%)	3	2	
GGT (IU/L) ^a^		52 (6–208)	92 (21–315)	0.342
AST (IU/L) ^a^		85 (22–517)	161 (15–392)	0.139
ALT (IU/L) ^a^		72 (12–268)	97 (13–201)	0.623
Bilirubin (μmoL/L) ^a^		11 (3–37)	12 (4–26)	0.695
ITU stay (days)		4 (1–28)	4 (1–10)	0.168
Inotrop support (Y/N)		19/16	6/6	1
Functional WIT (min)		NA	21 (9–33)	NA
CIT (min)		504 (210–840)	457 (270–720)	0.212
**Recipient**		**DBD (*n* = 35)**	**DCD (*n* = 12)**	***p*-Value ^b^**
Age (years)		44 (20–65)	54 (46–70)	0.029
Gender (female/male)		13/22	5/7	1
BMI (kg/m^2^)		25.8 (18.4–34.6)	27.3 (22.1–35.8)	0.277
MELD Score		14.3 (2–34)	10.7 (4–18)	0.208
UKELD Score		53.3 (40–77)	51.3 (44–61)	0.571
ALD		9	3	NA
PSC		5	0	
HCV		1	2	
HCC		1	2	
PHCC		2	1	
Others ^d^		17	4	
AST (IU/L) ^a^		480 (10–7485)	613 (18–5307)	0.494
Bilirubin day 7 (μmoL/L)		56 (7–258)	52 (12–103)	0.772
INR day 7		1.04 (0.85–1.21)	1.06 (0.92–1.3)	0.909
EAD/EGF		6/29	4/8	0.251
Censored/Dead ^c^		22/3	7/2	NA

DBD, donation after brain death; DCD, donation after circulatory death; GGT, gamma-glutamyl transferase; AST, aspartate aminotransferase; ALT, alanine aminotransferase; ITU, intensive therapy unit; WIT, warm ischaemia time; CIT, cold ischaemia time; BMI, body mass index; MELD, model for end-stage liver disease; UKELD, United Kingdom model for end-stage liver disease; ALD, alcoholic liver disease; PSC, primary sclerosing cholangitis; HCV, hepatitis C virus; HCC, hepatocellular carcinoma; PHCC, post hepatitis C cirrhosis; INR, international normalised ratio; EAD, early allograft dysfunction; EGF, early graft function. Continuous values are expressed as means (minimum–maximum); NA, not applicable. ^a^ Tested on the day of operation, ^b^ Mann–Whitney test (two-sided) or Fisher exact test (two-sided), ^c^ survival information was collected for 34 recipients, ^d^ other indications of liver transplantation include acute/chronic Wilson’s disease, metabolic disease, cholestatic disease, cryptogenic cirrhosis, polycystic disease, primary biliary cirrhosis, autoimmune cirrhosis, Alagille syndrome, hepatic malignancies, congenital biliary disease and unknow.

**Table 2 jcm-09-00711-t002:** ROC analysis for five annotated metabolites and five donor clinical parameters at pre-transplant for the prediction of EGF.

Indicators	AUC	Accuracy	Sensitivity	Specificity
adenine/urate + AMP/urate + hypoxanthine/urate + ALT	0.84	0.68	0.65	0.80
adenine/urate + adenosine/urate + AMP/urate + hypoxanthine/urate	0.80	0.70	0.65	0.90
AMP/urate	0.75	0.66	0.62	0.80
GGT + bilirubin + AST + ALT + age + steatosis status	0.71	0.57	0.57	0.60
adenine/urate	0.70	0.64	0.60	0.80
hypoxanthine/urate	0.68	0.53	0.51	0.60
bilirubin	0.65	0.68	0.67	0.70
AST	0.63	0.51	0.50	0.70
adenosine/urate	0.62	0.53	0.49	0.70
ALT	0.59	0.36	0.27	0.70
steatosis status	0.55	0.49	0.46	0.60
age	0.55	0.45	0.38	0.70
GGT	0.47	0.79	1	0

ROC, receiver operating characteristic; AUC, area under the curve; AMP, adenosine monophosphate.

**Table 3 jcm-09-00711-t003:** Partial correlation analysis (Pearson’s correlation, adjusting for patient age) between the levels of five selected metabolites and those of liver enzymes; *p*-values were represented as *q*-values after applying Benjamini–Hochberg correction; * *p* or *q* < 0.05, ** *p* or *q* < 0.01.

Metabolites		AST	Bilirubin	GGT
**Adenine**	Coefficient	−0.045	−0.122	−0.134
	*p*-value	0.968	0.321	0.275
	*q*-value	0.968	0.482	0.825
Adenosine	Coefficient	−0.005	−0.274	−0.084
	*p*-value	0.967	0.024 *	0.496
	*q*-value	0.967	0.072	0.744
AMP	Coefficient	−0.009	−0.097	−0.106
	*p*-value	0.945	0.430	0.390
	*q*-value	0.945	0.645	1
Hypoxanthine	Coefficient	−0.189	−0.320	−0.039
	*p*-value	0.122	0.008 **	0.752
	*q*-value	0.183	**0.024 ***	0.752
Urate	Coefficient	0.042	−0.019	0.204
	*p*-value	0.733	0.875	0.095
	*q*-value	1	0.875	0.285

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
