# Peer review of "Deregulation of the Purine Pathway in Pre-Transplant Liver Biopsies Is Associated with Graft Function and Survival after Transplantation"

_jcm, 2020, doi:10.3390/jcm9030711_

Round 1
Reviewer 1 Report
The current is an original article exploring the purine metabolites as predictors of early graft function and survival after liver transplantation from DBD and DCD. Authors conclude that a panel of purine metabolites in association with ALT is and effective predictive factor of EAD.
The manuscript is well-written and the topic is novel and very interesting. The methodology is well detailed; however it is not mentioned how patients were selected. Since all LT recipients were enrolled in a single center the selection criteria should be defined. In the study population, there is a small sample of patient who develop EAD after LT and mainly in the DCD cohort.
Despite no differences in the metabolite ratio has been found for the group of patients with steatosis, it is questionable if it depends from the small number of patients with moderate steatosis.
Since liver biopsies have been performed before and after graft reperfusion and differences have been found between metabolites levels, authors should comment on the different results and their implications on outcomes in the discussion.
Translating the dosage of purine metabolism in clinical practice is not so immediate, since it requires liver biopsies and specific tests, therefore a practical proposal for a real-life use of such predictive factors would add useful tools for its applicability.
Author Response
Reviewer 1 expressed a number of recommendations that we have addressed in this letter:
Reviewer comment:
The methodology is well detailed; however it is not mentioned how patients were selected. Since all LT recipients were enrolled in a single center the selection criteria should be defined.
Our response:
We thank the reviewer for pointing out that the description of selection criteria for recipients were not clearly clarified.
Changes to manuscript:
Page 2, 5th paragraph, lines 82-85: They were also presented with a similar severity of liver disease, represented by Model for End Stage Liver Disease (MELD) scores assessed at time of listing for LT. DCD donor liver grafts were randomly selected from transplants performed from August 2011 to August 2014 and all graft were matched with DBD grafts performed in the same period.
Reviewer comment:
In the study population, there is a small sample of patient who develop EAD after LT and mainly in the DCD cohort.
Our response:
We agree that the sample size of patients who develop EAD after LT is not big (10 out of 47), with the percentage of 21.3%. According to the demographics listed in Table 1, for the total of 10 patients who develop EAD, 6 were in the DBD cohort (17.1%) and 4 in the DCD cohort (33.3%).
Reviewer comment:
Despite no differences in the metabolite ratio has been found for the group of patients with steatosis, it is questionable if it depends from the small number of patients with moderate steatosis.
Our response:
The comparison of metabolite levels between normal and steatotic groups was conducted by allocating no steatotic donors (N=21) as normal group, mild and moderate steatotic donors (N = 26) as steatotic groups.
Changes to manuscript:
Page 4, 3rd paragraph, line 138-140: The levels of selected metabolite ratios between normal (no steatotic, n=21) and steatotic (mild and moderate steatotic, n=26) groups were also investigated.
Reviewer comment:
Since liver biopsies have been performed before and after graft reperfusion and differences have been found between metabolites levels, authors should comment on the different results and their implications on outcomes in the discussion.
Our response:
We agree with the reviewer that this is a limitation of our study with implications on outcomes and validation of results.
Changes to manuscript:
Page 10, 2nd paragraph, line 301-303: Also, a limitation of this study is that the first biopsy was taken before reperfusion, for optimal results a biopsy in the donor should also be included in the study design.
Reviewer comment:
Translating the dosage of purine metabolism in clinical practice is not so immediate, since it requires liver biopsies and specific tests, therefore a practical proposal for a real-life use of such predictive factors would add useful tools for its applicability.
Our response:
Agree and it was proposed in the discussion part of the manuscript (page 10, 2nd paragraph, line 303-305) that “To translate to the clinics and minimise the turnaround time of this panel (TAT), this test could be performed intraoperation, using available technology like rapid evaporative ionisation mass spectrometry (REIMS)”.
We want to thank the reviewer for a revising our manuscript and suggesting useful improvements.
Yours sincerely,
Dr. Cristina Legido-Quigley and Mr Wayel Jassem

Reviewer 2 Report
The manuscript by Xi et al. from the King's College is interesting, novel, and of clinical impact.
Therefore, with minor modification, this manuscript may be accepted for publication in the Journal.
The authors should address the following concerns:
1. TPMT-mutation analysis should be performed investigating purine metabolism in patients prior liver transplantation. Was this examination/test done to determine slow and fast metaboliser?
2. Do the authors expect different results in patients with different genetic background (SNP's)?
3. Can this panel of small molecules be performed bed-site? How long do we need to get the result?
4. Were protocoll biopsies post-transplantation performed in all recipients to investigate liver function?
Author Response
Reviewer 2 expressed a number of concerns that we have addressed in this letter:
Reviewer comment:
TPMT-mutation analysis should be performed investigating purine metabolism in patients prior liver transplantation. Was this examination/test done to determine slow and fast metaboliser?
Our response:
We thank the reviewer for a very interesting suggestion. This assay is not normally carried out in the clinical transplant setting; however, it would be a useful assay in a clinical trial that is being organised.
Reviewer comment:
Do the authors expect different results in patients with different genetic background (SNP's)?
Our response:
We thank the reviewer for raising this appealing question, and we are keen to perform transcriptomics and GWAS to test the genetic background, for this study we didn’t have the funds available to us.
Reviewer comment:
Can this panel of small molecules be performed bed-site? How long do we need to get the result?
Our response:
Yes and it was proposed in the discussion part of the manuscript (page 10, 2nd paragraph, line 303-305) that “To translate to the clinics and minimise the turnaround time of this panel (TAT), this test could be performed intraoperation, using available technology like rapid evaporative ionisation mass spectrometry (REIMS)”.
We are also organising this when the liver is in the donor and in a normothermic perfusion machine.
Reviewer comment:
Were protocoll biopsies post-transplantation performed in all recipients to investigate liver function?
Our response:
Yes, the routine-biopsies post-transplantation liver function were investigated in our Institute at King’s College Hospital.
We want to thank the reviewer for the revision and for pointing out interesting molecular avenues for us to investigate further at a systems level.
Yours sincerely,
Dr. Cristina Legido-Quigley and Mr Wayel Jassem
